# Structure and Texture Characteristics of Novel Snacks Expanded by Various Methods

**DOI:** 10.3390/ma16041541

**Published:** 2023-02-12

**Authors:** Katarzyna Lisiecka, Agnieszka Wójtowicz, Katarzyna Samborska, Marcin Mitrus, Tomasz Oniszczuk, Maciej Combrzyński, Jakub Soja, Piotr Lewko, Kamila Kasprzak Drozd, Anna Oniszczuk

**Affiliations:** 1Department of Biochemistry and Food Chemistry, University of Life Sciences in Lublin, Skromna 8, 20-704 Lublin, Poland; 2Department of Thermal Technology and Food Process Engineering, University of Life Sciences in Lublin, Głęboka 31, 20-612 Lublin, Poland; 3Department of Food Engineering and Process Management, Institute of Food Sciences, Warsaw University of Life Sciences—SGGW, Nowoursynowska 159C, 02-776 Warsaw, Poland; 4Department of Inorganic Chemistry, Medical University of Lublin, Chodźki 4a, 20-093 Lublin, Poland

**Keywords:** extruded snacks, carrot, expansion methods, morphometric parameters, texture

## Abstract

The aim of this work was to evaluate the structure of novel potato-based snack foods supplemented with various levels of fresh carrot pulp by using X-ray micro-computed tomography, texture profile, and sensory analysis. Three different methods of extruded snack pellets expansion were used to obtain ready-to-eat crisps: deep-fat frying, microwave, and hot-air toasting. The obtained results revealed that the pellets expansion method affected the porosity, size of pores and wall thickness, texture properties, and notes of sensory analyses of the obtained crisps. Deep-fat frying had a similar influence to microwave heating on ready-to-eat crisps properties, and both methods were significantly different in comparison to hot-air toasting. Crisps based on snack pellets supplemented with the addition of fresh carrot pulp in the amount of 10 to 30% expansion through hot-air heating showed unsatisfactory expansion and texture, but it is highly advisable to use deep-fat frying and microwave heating to achieve attractive potato-carrot crisps.

## 1. Introduction

Snack pellets requiring additional expansion after extrusion play an important role in the food market, in addition to snacks intended for direct consumption. These types of extruded snack pellets (known as non-expanded half-products) have to be converted into finished ready-to-eat (RTE) products by a separate expansion process [1,2]. Common methods of pellets expansion are frying in hot oil, microwaving, or puffing in hot air [3]. Deep-fat frying is based on the immersion and cooking of foods in a hot oil for a short time to obtain specific properties such as color, taste, flavor, crispness, and desired texture. During frying, the water present in foodstuffs evaporates from the product and the oil replaces it in situ. During expansion by microwave heating, the water present in the product undergoes heating inside the pellets, which generates bursting forces in the form of superheated steam. The accumulated steam creates local high-pressure, influencing the starch-based matrix and causing multidirectional expansion [4]. Additionally, the structure of the microwave-roasted sunflower seeds, for example, suggests microwaving can ensure effective protection against hydrolytic rancidity and deterioration of primary oxidation compounds, affecting its shelf life and introducing off-flavor and odor [5]. Hot-air heating or toasting, alongside microwave heating, is another suitable expansion method that enables the achievement of low-fat or fat-free fit snacks.

Among indirectly expanded snacks, the most common recipes are based on potato derivatives; however, to increase the nutritional value and sensory attractiveness of RTE snacks, supplementation with healthy components is possible. The pellets production process requires moistening the blend of dry components to the level of 25–36% water content and extrusion-cooking, and shaping and drying are the following steps. Using ground fresh vegetables as pulp during pellets production is one of the methods that can help save the technological water, which has a global meaning. Application of at least 5–10% of vegetable pulp significantly reduces the water amount needed for pellets extrusion-cooking, and if 30–40% of fresh pulp is applied in the recipe, any additional water is needed for processing [1]. Among the vegetables, carrot deserves a special attention due to the specific color, as well as high dietary fiber and carotenoids content. The carrot is characterized by proven pro-health activities, such as preventing cancer, reducing cardiovascular disease, lowering cholesterol level, etc. [6]. However, despite taking the healthiest additives in the composition, the selection of additives should be considered experimentally and the properties of such products should be tested both for processing stability and consumer acceptance. Among favorable features of extruded snacks, structural properties are one of the most important for consumer acceptability.

X-ray micro-computed tomography (mCT) seems to be an appropriate technique to evaluate snack food structure, because typical methods used to obtain information on food microstructure are mostly invasive and entail sample preparation, while mCT is a non-destructive and non-invasive way to obtain 3D imaging pictures and to generate several results that could be used for calculations suitable to describe the food structure [7]. This method could be applied in food sector research, as studying the properties of meat [8], dairy products [9], apples [10], vegetables [11], dried red bell peppers [12], freeze-dried strawberries [13], baked products [14], cereals [15], or extrudates obtained by extrusion-cooking such as ready-to-eat mealworms-supplemented snacks [16].

The aim of this study was to investigate the influence of different methods of expansion on selected properties of snacks by using the mCT technique, instrumental texture, and sensory analysis. The selected features of a new type of potato-based pellets and RTE snacks were supplemented with the addition of 10%, 20%, and 30% of fresh carrot pulp after expansion by such methods as deep-oil frying, microwave heating, and hot-air toasting. It is worth emphasizing that to our best knowledge, no previous studies exist that deal with such types of extrudates.

## 2. Materials and Methods

### 2.1. Materials

Potato flakes (PF), potato grits (PG), and potato starch (PS) were supplied by Pol-Foods Sp. z o.o. (Prostki, Poland), and the moisture content of raw materials was as follows: 9.3%, 7.5%, and 12.9%, respectively. Carrot (with a moisture content of 87.1%) was bought at a local market (Lublin, Poland).

### 2.2. Preparation of Recipes

Carrots were washed with tap water, dried with paper towels, and ground with a laboratory knife mill LMN-100—(TESTCHEM Sp. z o.o. Radlin, Poland) with 0.4 mm sieve openings to achieve a pulp with particle size below 400 μm. The recipes of the mixtures were established as a result of preliminary research. The basic recipe was based on potato flakes, potato grits, and potato starch at a ratio of 25:25:50. In tested blends supplemented with fresh carrot pulp (FCP), 10%, 20%, and 30% of potato starch were replaced with the appropriate amount (*w*/*w*) of FCP. All components were mixed together in a laboratory ribbon mixer for 6 min and additional water (if needed) was added to moisten the blends to a final moisture content of 33%, as recommended by Lisiecka and Wójtowicz [1,17].

### 2.3. Pellets Extrusion-Cooking

Extrusion-cooking of snack pellets was performed by using a single screw extruder type TS-45 (ZMCh Metalchem, Gliwice, Poland) with a plasticizing unit configuration of L/D = 18, equipped with a glycol cooling installation with the length of 6 D. The process parameters were selected experimentally on the basis of preliminary tests. The temperature range in individual extruder sections was as follows: dosing section, 80–85 °C; plasticizing section, 90–105 °C; cooling section, 50–65 °C; forming die, 60–75 °C. Processing of the tested blends was performed at 80 rpm screw speed. The obtained extrudates were shaped as a dough strips using a 30 × 0.4 mm flat forming die, and final pellets were cut to approx. 30 × 30 mm pieces and dried in a laboratory shelf dryer at 40 °C for 10 h until the final moisture content was 11%. The dried snack pellets were kept in tightly closed foil packages before further tests.

### 2.4. Expansion Process

The snack pellets expansion procedure was carried out in three different ways. The conditions of the expansion methods were determined experimentally in all ways. The first method was frying, with the pellets expanded by frying in hot oil at 180 °C for 15 s with the excess of oil drained from the surface with paper towels. The second method was microwave treatment, using a microwave oven (AVM-914/WH Philips/Whirlpool, Norrkoeping, Sweden) with a power of 800 W for 40 s duration. The third method was expansion by the application of an air fryer (FTL1004, Kalorik, Sint-Genesius-Rode, Belgium), with the power of 1000 W at 170 °C, for 210 s duration. All expansion parameters were preselected experimentally. The obtained expanded crisps were stored at room temperature in closed plastic bags before analyses.

### 2.5. X-ray Microcomputed Tomography

The internal microstructure of pellets and RTE snacks was characterized in three dimensions (3D) at a micron-level spatial resolution by non-destructive X-ray microcomputed tomography (mCT). Four types of samples were analyzed: snack pellets (P), crisps expanded by frying in hot oil (F), crisps expanded by microwave heating (M), and crisps expanded by air toasting (A). For further description, the variants were also marked with numbers indicating the level of FCP addition: 0%, 10%, 20%, 30%. Internal microstructure of two individual samples (a piece with a size of approx. 1.5 × 2.5 cm cut from a randomly chosen whole snack piece) of each processing variant was characterized.

#### 2.5.1. Image Acquisition and Reconstruction

Sample scans were performed on a SkyScan 1272 system (Bruker microCT, Kontich, Belgium) operated at 45 keV source voltage and 201 μA current with the following parameters: image pixel resolution of 25.0 μm, rotations by 0.3° steps over a total of 180°, averaging 4 frames. A stack of about 628 projection raw images (1008 × 1008 pixels) was obtained. The scan duration for each sample was 45 min.

#### 2.5.2. Reconstruction

The raw projection images were loaded in NRecon 1.6.3.2 software (Bruker, Kontich, Belgium) to reconstruct a virtual cross-section of the sample in the image range 0–0.07. The images were corrected for rings and beam hardening (common artifacts in mCT images). Misalignment compensation was performed separately and automatically for each sample.

#### 2.5.3. Visualization

The 3D visualization was performed in CTvox software (Bruker, Kontich, Belgium). A custom palette was created to visualize the color of samples.

#### 2.5.4. Data Analysis

Reconstructed data sets were loaded in CTAnn software (Bruker, Kontich, Belgium) and binarized in a threshold range 35–255. After the adjustment of the region of interest by a shrink-wrap internal plugin in custom processing, 3D analysis was performed to derive the total porosity (TP—the volume of all open plus closed pores), closed porosity (CP), and the thickness of the walls. Wall thickness is the thickness of the layers of solid material located between pores and can be described as the distance between one surface of the solid and its opposite surface. Afterwards, the threshold was reversed by bitwise operations plug-in, which was again followed by 3D analysis, resulting in the determination of pore size. Subsequently, the 3D morphometric parameters were derived as follows: the object surface/volume ratio (OS/V), the object surface density (OSD—the ratio of surface area to total volume measured), and the Euler number (EN) as an indicator of connectedness of a 3D complex structure.

### 2.6. Texture Profile Analysis of Expanded Snacks

Texture measurements were made using a Zwick/Roell BDOFB0.5TH (Zwick GmbH and Co., Ulm, Germany) universal testing machine equipped with a five-blade Kramer shear cell for a single expanded crisp in five replications. The test speed of the working head was set at 3.3 mm/s. For all expanded crisps texture profile included hardness (H), crispiness (CR), and fracturability (F). H was evaluated as the maximum force in the test cycle. CR described the difference between the first significant force and the first minimum force following after the first significant force during the test. F was the first significant force drop in the test cycle [17].

### 2.7. Sensory Analysis of Expanded Crisps

The team included 15 semi-trained members who evaluated the shape, color, flavor, taste, crispness, and overall quality of RTE crisps on a scale from 1 (“extremely dislike”) to 9 (“extremely like”) points, wherein 5 points was defined as “neither like-nor dislike” [18]. Expanded crisps were regarded as acceptable when scored above 5, according to Wójtowicz and coworkers [18]. The panelists were aware of the purpose of sensory analysis. The test was carried out in a room with white light at a temperature of 20 °C.

### 2.8. Statistical Analysis

The obtained results underwent statistical analysis using Statistica 13.3 software (StatSoft, Tulsa, OK, USA). Data were subjected to one-way analysis of variance (ANOVA) by significant differences, with Tukey’s post hoc test applied to compare means at the α = 0.05 significance level. Principal component analysis (PCA) was also performed for the tested microstructure, texture, and sensory profile of expanded crisps supplemented with FCP and depending on the expansion method applied. A dendrogram plot was then prepared to analyze clustered data.

## 3. Results and Discussion

Figure 1 presents the 3D visualization of snacks pellets and RTE crisps expanded by frying or by microwave heating or hot-air toasting. For all tested samples, it was found that the external structure did not vary significantly due to the increasing level of FCP. In contrast, differences were visible between the methods of expansion applied. The unexpanded snack pellets showed the most homogenous glassy structure (P0, P10, P20, P30), while crisps fried in deep oil were characterized by uniformly porous structure (F0, F10, F20, F30) and were less compacted than crisps expanded using microwave heating (M0, M10, M20, M30). In the case of hot-air-expanded crisps (A0-A30), the smooth surface structure observed was more similar to the unprocessed pellets, due to a less intensive expansion than that of fried or microwaved crisps.

The internal structure was assessed via cross-sections of unexpanded snack pellets and expanded crisps (Figure 2). Herein, the presence of pores was developed and visible after expansion by all applied methods in comparison to unprocessed pellets. However, increased levels of fresh carrot additive reduced the number and pore size of crisps expanded by all methods applied. Lisiecka and Wójtowicz [19] also noticed that an increased level of fresh beetroot pulp as a supplementary ingredient in snack pellets recipe created a denser internal structure of deep-oil fried crisps.

When comparing the effect of the expansion method of snack pellets supplemented with similar levels of FCP, the fried crisps showed the most porous internal structure. Furthermore, their internal cell walls appeared to be the thinnest when compared to that of microwave expanded or hot-air-expanded crisps. The significantly weakest expansion was noted in the case of A30 sample with the highest vegetable pulp content as expanded by hot-air toasting. This indicates the limitation level at 30% in the snack pellets recipe in this type of fat-free expansion method.

### 3.1. Morphometric Characteristics of Snack Pellets and Crisps Derived by mCT

The derived morphometric parameters of snack pellets and expanded crisps supplemented with the addition of FCP are presented in Table 1. As it was noted, the total porosity varied from 2.30% to 74.50% and was dependent on the sample type (pellet/crisp), the expansion method, and the addition of FCP. The highest total porosity was observed in sample M0, indicating that the greatest expansion was obtained in the control sample without fresh carrot pulp addition after the application of microwave expansion, as shown in Figure 2. Here, mCT cross-section images revealed the highly porous internal structure of crisps, which was confirmed by the results of porosity calculations. Low TP was observed for all unexpanded snack pellets, which is natural for these kinds of extrudates, and it ranged from 2.30% (pellets with the maximum level of FCP) to 5.40% (pellets with 20% addition of vegetable carrot pulp). Among the snacks, the lowest total porosity displayed crisps supplemented with 30% of FCP expanded by hot-air toasting (A30 sample).

Closed porosity varied from 0.08% for M30 crisps to 2.21% for A30 sample. The microwave expanded crisps with the highest FCP content characterized by the maximum value of CP were significantly different from the other samples. In related work, Gondek et al. [20] observed that the closed porosity of whole-grain rye flour crispbread obtained by different combinations of extrusion parameters did not exceed 0.78%, suggesting that the majority of the pore space was interconnected in a 3D matrix during production.

The Euler number describes the number of objects minus the number of redundant connections or loops and the number of completely enclosed cavities or holes [21]. A high value of the Euler number indicates a poorly connected structure, while low values indicate a properly (well) connected structure. In this study, all the unexpanded snack pellets (P0-P30) as well as the hot-air toasted crisps with the highest FCP content (A30) showed positive values of the Euler number, ranging between 406 and 12,113 (Table 1). Other types of expanded crisps showed negative values ranged from −32,609 to −122,431. The Euler number thus indicated an expansion as a result of thermal processing: crisps with a porosity above 40% had an absolute Euler number greater than 30,000. Additionally, the sample M0 with the highest total porosity had the highest absolute value of the Euler number.

The total porosity of all expanded samples decreased with the increased addition of FCP (Table 1). Similarly, the results presented by Azzollini et al. [16] showed that the addition of 20% of ground mealworms during the extrusion-cooking of ready-to-eat snacks progressively reduced the size of extruded snacks, the overall porosity, and the pore size. This is likely due to a reduction in the amount of converted starch as well as an increase in protein and lipid content after enrichment. Microwave-expanded control potato crisps without vegetable addition (M0) showed the highest surface area to volume ratio (OS/V = 39.01 mm^−1^), indicating a fine, highly branched structure. The lowest among others value of OS/V (9.11 mm^−1^) obtained for hot-air toasted crisps with the highest addition of FCP (A30) was statistically more similar to the values obtained for the unexpanded snack pellets than for other expanded crisps. The object surface density, which may indicate the ratio of aeration to the sample surface, ranged from 2.63 mm^−1^ (P0) to 12.83 mm^−1^ (M30). In the case of F and M expanded crisps, there was a tendency to increase the OSD value with increasing addition of FCP. This suggests that this additive under frying and microwaving increased surface aeration. This phenomenon can be observed in the visualization images shown in Figure 1, especially in the case of RTE crisps expanded by the microwave heating. In contrast, crisps expanded by hot-air toasting did not show this effect.

Table 2 presents the correlation matrix of selected parameters of the expanded crisps. The total porosity was significantly negatively correlated with closed porosity (−0.94) and the Euler number (−0.90), and positively correlated with the object surface to volume ratio (0.89).

In the case of 3D rice pellets, it was noticed that the porosity ranged from 79.1% (for pellets made of a blend moistened to 29% extruded at 135 °C) to 83.5% (for pellets made of a mixture moistened to 31% extruded at 110 °C ) during their expansion using 800 W microwaving for 50 s. However, it was noted that in statistical terms, the pellet extrusion conditions had no statistical significance in the case of porosity [22]. In the typical potato pellets expanded by deep-frying, it has been observed that the inclusion of fiber inhibits the expansion and formation of a porous structure, which is associated with the lowering amount of starch responsible for expansion process [23]. In the case of the porosity of deep-fried products, the main factors analyzed were the effects of time, temperature, drying, the properties of the material, or type of oil used [24]. The closed porosity of the expanded crisps was significantly positively correlated with the Euler number (0.87) and negatively with the object surface/volume ratio (−0.88) and the object surface density (−0.58). The Euler number was negatively correlated with object surface/volume ratio (−0.86). At the same time, the object surface/volume ratio values were positively correlated with the object surface density (0.57).

### 3.2. Distribution of Pore Size and Walls Thickness in Snack Pellets and Expanded Crisps Derived by mCT

The curves of pore size and wall thickness distribution are shown in Figure 3. The pore-size distribution of unexpanded snack pellets ranged from 50 to 550 μm, and as the FCP content in the recipe increased, it became narrower and shifted towards lower values. Nevertheless, pore sizes between 50 and 100 μm were dominant. The pore-size distribution in the expanded crisps was quite different to that of the unexpanded snack pellets. A wider range was observed here, indicating the presence of pores of different sizes, which also differed in the experimental variants used. The widest range of crisps’ pore size was observed in the fried control samples (F0), and when a small amount of FCP additive was used (F10), it varied from 50 to 2200 μm. The narrowest range was observed in hot-air-expanded samples supplemented with the highest FCP addition (A30). It was between 50 to 1150 μm. However, for this variant, the size distribution above 1150 μm did not exceed 7% in relation to the total distribution. For comparison, Alam and Takhar [25] reported that the 1.65 mm thick raw potato discs had a pore-size range from 4 to 100 μm; however, after frying, it ranged from 10 to 900 μm.

The phenomenon of pore-size increase after the expansion process in the case of frying method is related to the evaporation of water both from the surface and from the central part of the product. Some of this vapor may become trapped inside the porous matrix, and then expand, thus disrupting the pore structure. Zhang, Liu, and Fan [26], while frying potato slices 2 mm thick and 22 mm in diameter in palm oil at 180 °C for 20–160 s, observed that the pore diameter of 100–200 μm was dependent on frying time and increased from 5 to 20% with increasing frying time. They also noticed that the smallest pore sizes (0.3–10 μm) disappeared as the frying continued due to the breakage of the pores and fusion. Kawas and Moreira [27], while studying the structure of tortilla chips, also speculated that the formation of larger pores could be attributed to steam expansion and bursting of the pores. Prawiranto et al. [28], studying the effect of drying methods on changes in the microstructure of fruit, observed the development of layers with increased and reduced porosity. In this study, it was noticed that all drying methods caused an increase in the pore diameter and were correlated with the layer with increased porosity. The fact that several small pores have merged into one with a larger diameter explains the increase in the average pore diameter.

There are also reports in the literature where the influence of pore size and wall thickness is attributed to the increasing level of the additive supplementing the products. In the case of fried snacks enriched with fresh beetroot pulp in the amount up to 30%, Lisiecka and Wójtowicz [19] observed that after frying the supplemented pellets, an increased level of fresh beetroot resulted in a denser internal structure. In contrast, the cell walls of the pores visible in the snacks were characterized by a thinner and more delicate structure with a smaller number of large pores. ranging from 4 to 100 μm, but after frying, it ranged from 10 to 900 μm.

In ready-to-eat snacks with increased content of mealworms from 10 to 20%, a decrease in mean pore size and an increase in wall thickness was accompanied by a reduced expansion. These effects were associated with an increased level of the additive in the recipe [16]. Similar trends were observed in the production of snacks made of barley flour and carrot pomace in the amount of 10%, 17.5%, and 25% [29], or corn snacks with the addition of fresh kale pulp up to 20% [30]. This phenomenon consists in the reduction in the amount of starch in the processed material. This occurs in plant supplements because the fibrous fraction leads to the destruction of the cellular structure of the starch matrix. The influence of the interaction between starch and protein from plant products or the increased moisture of the processed raw materials should also be taken into account.

The results showed that the percentage of pore size in snacks from 50 μm and 550 μm was positively correlated with the object surface density (0.60) and negatively correlated with the percentage of pore size from 550–1150 μm (−0.95) and over 1150 μm (−0.76). The percentage of the range from 550 to 1150 μm was positively correlated with the percentage of pore size above 1150 μm (0.57).

In the presented research, the range of pellet wall thickness was narrowed along with the increase in the addition of fresh carrot pulp. The most common thickness (over 50%) in the pellets was between 900 and 1350 µm. The expansion led to a reduction in the thickness of the walls. The numerous thick walls in unexpanded snack pellets and thin walls in expanded crisps are easily observable in the 3D visualizations and cross-sections (Figure 1 and Figure 2). The widest range of wall thickness distribution in expanded crisps was observed in the fried sample supplemented with 10% of FCP (F10). Here, the values ranged from 50 to 1150 μm. The narrowest range of pore-size distribution was observed in the samples expanded by hot-air toasting (A0 and A20) and ranged from 50 to 400 μm. The wall thickness above 400 μm did not exceed 14% in all other samples.

Prawiranto et al. [28] reported that, in the case of natural convection, the thickness of the layer with increased porosity decreased along with the extension of the drying time. This was due to the axial shrinkage of the upper part of the tissue. On the other hand, the layer of increased porosity increased with drying time in the process of forced convection. In the case of irradiation–convection, the increased reduced porosity was associated with high volumetric shrinkage, which induced structure densification.

The percentage of expanded crisp wall thickness ranging from 50 μm to 200 μm was positively correlated with the total porosity (0.84), the object surface/volume ratio (0.90), and the object surface density (0.72), and negatively correlated with the closed porosity (−0.89), the Euler number (−0.78), the percentage of the wall thickness of crisps ranging from 200 μm to 400 μm (−0.97) and above 400 μm (−0.60). Additionally, the percentage of crisp wall thickness in the range from 200 μm to 400 μm was negatively correlated with the total porosity (−0.93), the object surface/volume ratio (−0.91), and the object surface density (−0.64), and positively with the closed porosity (0.96) and the Euler number (0.83). The percentage of expanded crisp wall thickness above 400 μm was only correlated with the object surface density (−0.65).

The porous microstructure of fruit and vegetable chips is based on the architecture of the plant walls between the pores, and is responsible for the hardness and crunchiness of crispy products through the thickness and rigidity of the walls [31]. Peng et al. [32], in a study of carrot snacks subjected to osmotic treatment under various conditions before drying, found that thinner and concentrated walls lead to a greater crispness of the sample.

### 3.3. Results of Texture and Sensory Analysis Measurements of Snacks

Table 3 shows the texture profile and the results of the sensory analysis of the expanded snack foods. The crisps expanded by hot-air toasting with the highest content of FCP (A30) were characterized by the highest value of hardness and the lowest fracturability and crispiness. In contrast, the minimum hardness was observed for fried samples supplemented with the highest amount of carrot (F30), and the maximum fracturability and crispiness was noted for microwaved crisps without additive (M0). Moreover, there was a significant difference in fracturability between M0 and all hot-air-expanded samples. In the case of crispiness, a significant difference was observed between M0 and all hot-air-expanded samples as well as M30, F10, and F30 crisps. The hot-air-expanded crisps were significantly harder compared to other samples, except M20. The most attractive crisps were fried products without additives (F0), which received the best ratings for shape, flavor, taste, crispiness, and overall score. However, in terms of color, the F20 and F30 fried crisps were better than the others, but the differences were statistically insignificant. The crisps expanded in the hot air turned out to be the least attractive.

Statistical analysis (Table 2) showed that selected properties describing the texture of RTE crisps were correlated with morphometric parameters. However, the strongest correlation was between fracturability, crispness, and object surface/volume ratio (0.85 and 0.73, respectively), and between hardness and total porosity (−0.66). Fracturability and crispness were correlated with the percentage wall thickness of the crisps ranging from 50 to 200 μm (0.66 and 0.55, respectively). It was also observed that the percentage of crisp wall thickness ranging from 200 to 400 μm was negatively correlated with fracturability (−0.67) and crispness (−0.55), and positively with hardness (0.55). In the case of sensory analysis, it was noticed that among the morphometric and textural parameters, the strongest correlation is between total porosity and hardness according to shape (0.69 and −0.88), color (0.67 and −0.85), flavor (0.86 and −0.91), taste (0.69 and −0.89), crispness (0.74 and −0.87), and overall score (0.76 and −0.90), respectively. A very strong correlation was also observed between the parameters of the sensory analysis. The flavor was positively correlated with the percentage of wall thickness of expanded crisps ranging from 50 to 200 μm (0.59). There was also a negative correlation between the percentage walls thickness of RTE snacks ranging from 200 to 400 μm and flavor (−0.71), taste (−0.54), crispness (−0.58), and overall score (−0.58) (Table 2).

The results of statistical analysis showed that Instrumental assessment of the extruded products texture has been related to sensory evaluation, as confirmed by Anton and Luciano [33], and in snack foods, texture is of major importance, with crispness being a desirable attribute. In the case of snacks based on blends of maize and partially defatted soybean in an amount from 0 to 30%, it was observed that an increase in the hardness of the sample measured instrumentally resulted in a decrease in the overall acceptability of the product during sensory analysis [34]. Altan and coworkers [35,36] noted that reducing the porosity of the extrudate structure increases the snack hardness through the presence of fiber.

### 3.4. Principal Component Analysis and Hierarchical Cluster Analysis

Approximately 77% of the sample data variance was explained by the first two principal components: PC1 (55.59%) and PC2 (21.54%). PC1 was negatively correlated with the closed porosity, Euler number, hardness, percentage wall thickness of RTE snacks ranging from 200 to 400 μm, and positively with total porosity, object surface/volume ratio, fracturability, crispness, all parameters of the sensory analysis, and percentage of wall thickness of RTE snacks ranging from 50 to 200 μm. PC2 was negatively correlated with the object surface density and the percentage of crisps pore size between 50 and 550 μm, and positively correlated with the percentage of crisps pore size between 550 and 1150 μm and the percentage walls thickness of RTE crisps above 400 μm (Figure 4).

Placing the samples in the space of the first two factors showed that the samples were different from each other (Figure 5). Moreover, significant differences can be observed between clusters. The first cluster included sample A30 (with the addition of 30% FCP expanded by hot-air toasting). These crisps differed from other samples expanded by hot-air heating (cluster 2), hot-oil-fried, and microwave-heated RTE snacks (cluster 3), but were closer to the unexpanded pellets, meaning that the hot-air expansion method did not provide sufficient expansion.

From the chart in Figure 5a it can be concluded that the first factor up to 77% describes the variability of the system and determines the snack pellets expansion rate. Another possibility to analyze the cluster results is to generate a dendrogram graph where the formation of individual clusters can be observed. This confirmed the conclusions described above (Figure 5b).

## 4. Conclusions

The microstructure of the crisps expanded by various methods was significantly different from the unexpanded snack pellets. Regardless of the method used, the expansion increased the range of pore size distribution and decreased the wall thickness. The results showed different effects of the fresh carrot pulp application and expansion methods on the texture properties of crisps, with hot-air-expanded snacks being found to have the highest hardness. Due to the low scores in the sensory evaluation, it is not recommended to expand the pellets with the addition of carrot pulp in the range from 10 to 30% by hot-air heating under the proposed conditions. However, by using other methods of expansion, the vegetable content can be increased to 30% of fresh carrot pulp in potato-based recipe. In addition to the above, it was confirmed that the MicroCT method is a suitable method for assessing the internal structure and morphometric features of potato-based extrudates supplemented with FCP expanded by various methods.

## Figures and Tables

**Figure 1 materials-16-01541-f001:**
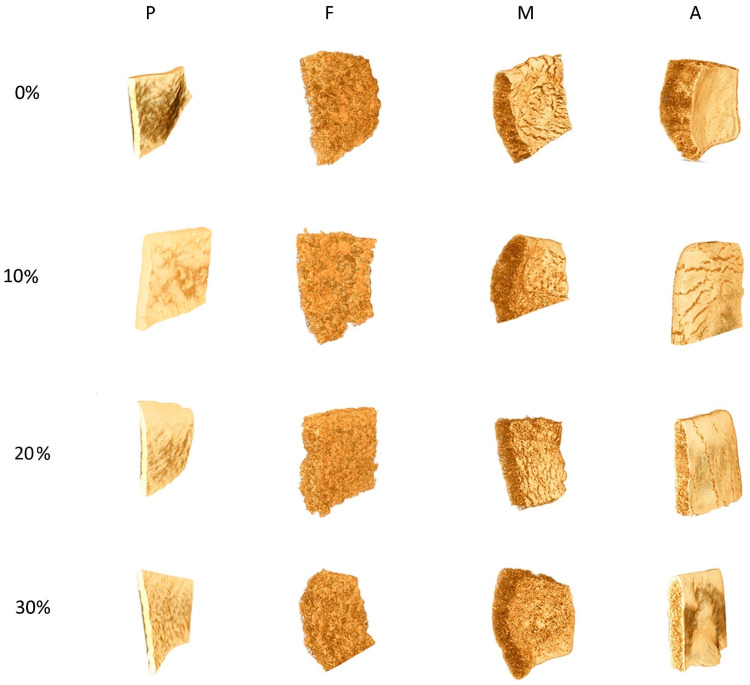
3D visualization of snack pellets and crisps expanded by selected methods: P—unexpanded pellets, F—crisps expanded by frying in deep oil, M—crisps expanded by microwaving, A—crisps expanded by hot-air toasting, 0–30 amount of fresh carrot pulp (%).

**Figure 2 materials-16-01541-f002:**
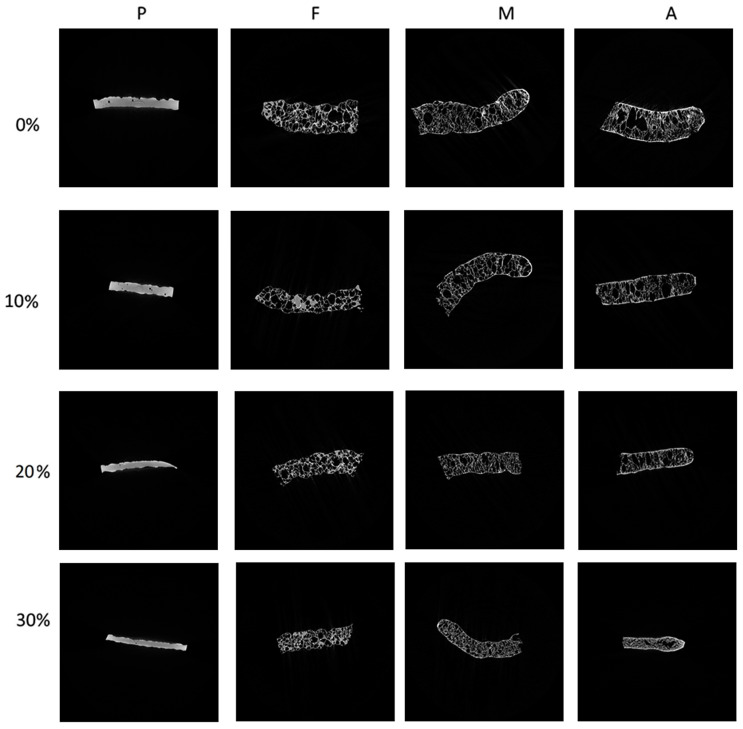
Cross-section of snack pellets and crisps expanded by selected methods: P—unexpanded pellets, F—crisps expanded by frying in deep oil, M—crisps expanded by microwaving, A—crisps expanded by hot-air toasting, 0–30 amount of fresh carrot pulp (%).

**Figure 3 materials-16-01541-f003:**
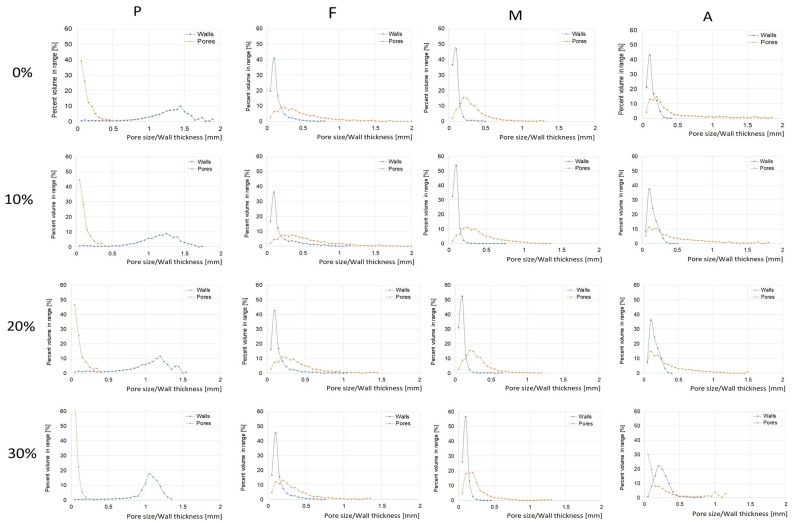
Pore size and wall thickness distribution in snack pellets and crisps expanded by selected methods: P—unexpanded pellets, F—crisps expanded by frying in deep oil, M—crisps expanded by microwave, A—crisps expanded by hot-air toasting, 0–30 amount of fresh carrot pulp (%).

**Figure 4 materials-16-01541-f004:**
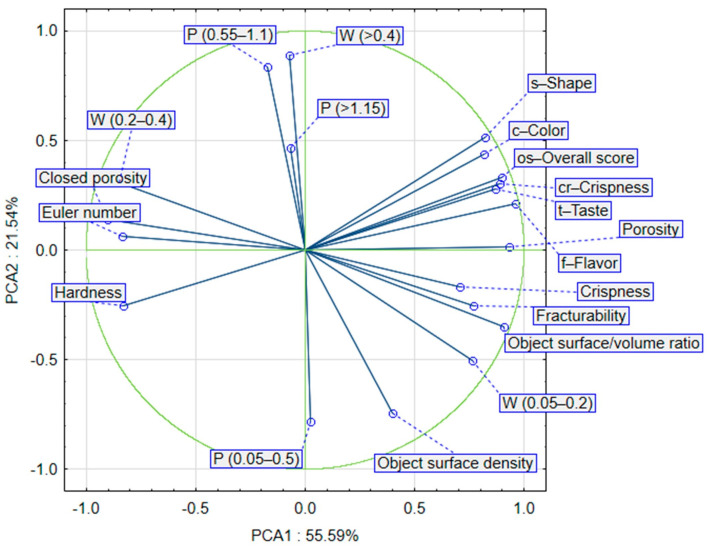
PCA analysis of snack pellets and expanded crisps features.

**Figure 5 materials-16-01541-f005:**
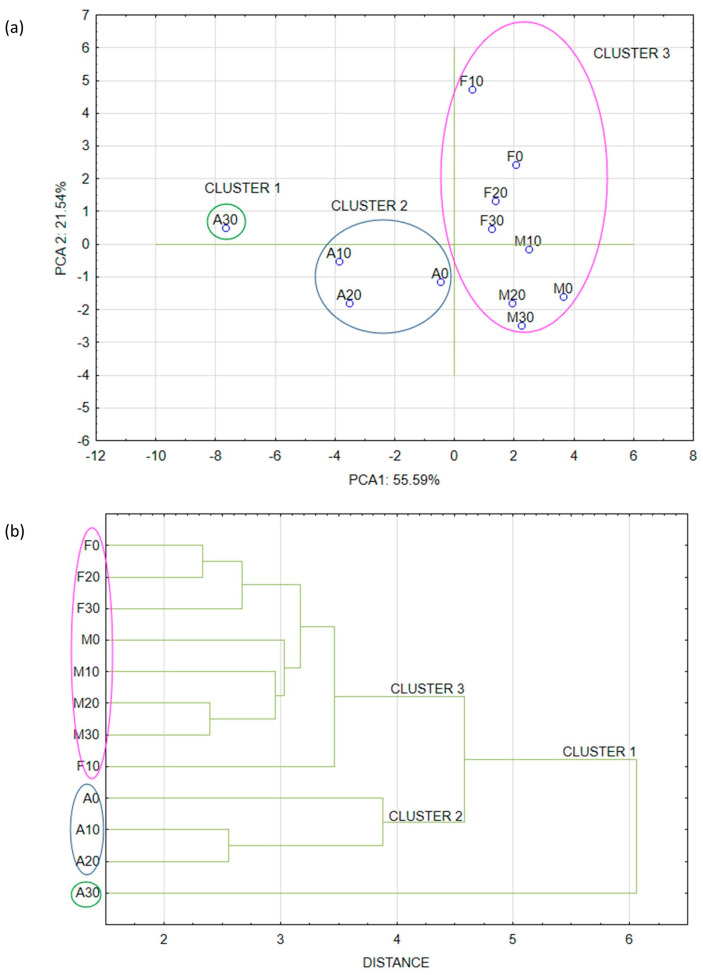
The crisps objects in space of first two major components by PCA (principal component analysis) (**a**), and the dendrogram obtained by HCA (hierarchical cluster analysis) application (**b**). F—crisps expanded by frying in deep oil, M—crisps expanded by microwave, A—crisps expanded by hot-air toasting, 0–30 amount of fresh carrot pulp (%); different colors of circles mean similar homogenous groups.

**Table 1 materials-16-01541-t001:** Selected morphometric parameters of snack pellets and expanded crisps with the addition of fresh carrot pulp.

TS	AA [%]	TP [%]	CP [%]	EN [-]	OS/VR [mm^−1^]	OSD [mm^−1^]
P	0	4.80 ± 1.03 ^a^	0.28 ± 0.05 ^a^	834.00 ± 482.00 ^cd^	2.76 ± 0.11 ^a^	2.63 ± 0.09 ^a^
	10	4.60 ± 2.64 ^a^	0.43 ± 0.07 ^a^	1830.50 ± 1269.50 ^cd^	2.80 ± 0.19 ^a^	2.67 ± 0.13 ^a^
	20	5.40 ± 0.15 ^a^	0.37 ± 0.11 ^a^	472.50 ± 352.50 ^cd^	3.54 ± 0.41 ^a^	3.35 ± 0.39 ^a^
	30	2.30 ± 0.10 ^a^	0.34 ± 0.07 ^a^	406.00 ± 312.00 ^cd^	3.02 ± 0.23 ^a^	2.95 ± 0.14 ^a^
F	0	67.94 ± 2.79 ^fgh^	0.41 ± 0.04 ^a^	−93,453.50 ± 23,511.50 ^ab^	27.13 ± 1.02 ^bcd^	8.68 ± 0.21 ^bcd^
	10	67.30 ± 4.40 ^fgh^	0.45 ± 0.09 ^a^	−80,732.50 ± 35,405.50 ^ab^	24.03 ± 2.11 ^bcd^	7.80 ± 0.06 ^bc^
	20	63.40 ± 2.67 ^ef^	0.40 ± 0.28 ^a^	−63,418.50 ± 14,763.50 ^abcd^	25.85 ± 2.19 ^bcd^	9.43 ± 0.31 ^cde^
	30	60.50 ± 0.66 ^de^	0.33 ± 0.12 ^a^	−57,831.50 ± 2329.50 ^abcd^	27.43 ± 1.41 ^de^	10.84 ± 0.43 ^e^
M	0	74.50 ± 1.49 ^h^	0.11 ± 0.09 ^a^	−12,2431.00 ± 8687.50 ^a^	39.01 ± 1.59 ^f^	9.93 ± 0.01 ^de^
	10	73.84 ± 0.60 ^h^	0.25 ± 0.17 ^a^	−104,558.50 ± 16,920.50 ^ab^	36.02 ± 0.07 ^ef^	9.42 ± 0.13 ^cde^
	20	69.90 ± 2.63 ^gh^	0.37 ± 0.30 ^a^	−69,596.50 ± 17,823.50 ^abc^	35.67 ± 1.71 ^ef^	10.70 ± 0.15 ^e^
	30	63.40 ± 2.43 ^ef^	0.08 ± 0.03 ^a^	−87,833.00 ± 19,973.00 ^ab^	35.08 ± 1.02 ^ef^	12.83 ± 0.23 ^f^
A	0	65.20 ± 3.95 ^fgh^	0.19 ± 0.0 ^a^	−115,091.00 ± 856.00 ^a^	30.09 ± 1.87 ^de^	10.42 ± 0.19 ^de^
	10	52.80 ± 3.49 ^cd^	0.92 ± 0.3 ^a^	−51,624.00 ± 7723.00 ^abcd^	20.48 ± 1.13 ^bc^	9.65 ± 0.03 ^cde^
	20	48.50 ± 0.30 ^c^	0.74 ± 0.15 ^a^	−32,609.00 ± 4792.00 ^bcd^	20.23 ± 0.12 ^b^	10.42 ± 0.02 ^de^
	30	17.20 ± 1.70 ^b^	2.21 ± 0.28 ^b^	12,113.00 ± 2273.00 ^d^	9.11 ± 1.38 ^a^	7.53 ± 1.04 ^b^

TS—type of sample; P—snack pellets, F—crisps expanded by frying in hot oil; M—crisps expanded by microwave; A—crisps expanded by air frying; AA—additive amount (fresh carrot pulp, %); TP—total porosity; CP—closed porosity; EN—Euler number; OS/V—object surface/volume ratio; OSD—object surface density; ^a–h^—means indicated with similar letters in columns do not differ significantly at α = 0.05.

**Table 2 materials-16-01541-t002:** Correlation matrix of selected parameters of expanded crisps.

	TP	CP	EN	OS/VR	OSD	FR	CR	H	S	C	F	T	C1	OSS	P1	P2	P3	W1	W2
CP	−0.94 *																		
EN	−0.91 *	0.87 *																	
OS/VR	0.89 *	−0.88 *	−0.86 *																
OSD	0.35	−0.58 *	−0.31	0.57 *															
FR	0.72 *	−0.61 *	−0.70 *	0.85 *	0.24														
CR	0.65 *	−0.54 *	−0.66 *	0.73 *	0.08	0.92 *													
H	−0.66 *	0.66 *	0.54 *	−0.61 *	−0.26	−0.45	−0.38												
S	0.69 *	−0.59 *	−0.55 *	0.54 *	−0.07	0.52 *	0.52 *	−0.88 *											
C	0.67 *	−0.64 *	−0.54 *	0.54 *	0.06	0.44	0.48	−0.85 *	0.95 *										
F	0.86 *	−0.80 *	−0.72 *	0.78 *	0.22	0.68 *	0.66 *	−0.91 *	0.93 *	0.92 *									
T	0.69 *	−0.64 *	−0.55 *	0.66 *	0.19	0.57 *	0.48	−0.89 *	0.93 *	0.92 *	0.92 *								
C1	0.74 *	−0.67 *	−0.60 *	0.66 *	0.16	0.59 *	0.54 *	−0.87 *	0.94 *	0.92 *	0.93 *	0.98 *							
OSS	0.76 *	−0.71 *	−0.65 *	0.67 *	0.16	0.53 *	0.50 *	−0.90 *	0.95 *	0.96 *	0.95 *	0.98 *	0.98 *						
PO1	−0.20	0.05	0.15	0.24	0.60	0.22	0.16	0.05	−0.26	−0.14	−0.08	0.02	−0.07	−0.09					
PO2	−0.00	0.18	0.06	−0.41	−0.77 *	−0.27	−0.20	0.01	0.21	0.06	−0.02	−0.08	−0.02	−0.01	−0.95 *				
PO3	−0.18	−0.16	−0.27	−0.16	−0.23	−0.36	−0.32	0.13	−0.01	0.00	−0.09	−0.16	−0.07	0.00	−0.76 *	0.57 *			
W1	0.84 *	−0.89 *	−0.78	0.90 *	0.72 *	0.66 *	0.55 *	−0.44	0.30	0.31	0.59 *	0.41	0.45	0.44	0.17	−0.38	−0.03		
W2	−0.93 *	0.96 *	0.83 *	−0.91 *	−0.64 *	−0.67 *	−0.55 *	0.55	−0.46	−0.46	−0.71 *	−0.54 *	−0.58 *	−0.58 *	−0.02	0.24	−0.07	−0.97 *	
W3	−0.11	0.22	0.21	−0.38	−0.65 *	−0.23	−0.20	−0.15	0.42	0.40	0.13	0.26	0.24	0.26	−0.53 *	0.62 *	0.29	−0.60 *	0.39

TP—total porosity; CP—closed porosity; EN—Euler number; OS/V—object surface/volume ratio; OSD—object surface density; FR—fracturability; Cr—crispness; H—hardness; S—shape; C—color; F—flavor; T—taste; C1—crispness of sensory analysis; OSS—overall sensory score; PO1—the percentage amount of the pore size crisps between 50 and 550 μm; PO2—the percentage amount of the pore size crisps between 550 and 1150 μm; PO3—the percentage amount of the pore size crisps above 11,500 μm; W1—the percentage amount of wall thickness of crisps in the range from 50 to 200 μm; W2—the percentage amount of wall thickness of crisps in the range from 200 to 400 μm; W3—the percentage amount of wall thickness of crisps above 400 μm; *—significant at α = 0.05.

**Table 3 materials-16-01541-t003:** The results of texture profile and sensory analysis of expanded snacks with the fresh carrot pulp addition.

TS	AA [%]	FR [N]	CR [N]	H [N]	Shape	Color	Flavor	Taste	Crispness	Overall Score
F	0	22.93 ± 4.83 ^abc^	22.00 ± 4.36 ^ab^	63.50 ± 2.41 ^ab^	7.67 ± 0.94 ^c^	7.50 ± 0.96 ^ab^	6.67 ± 1.49 ^a^	7.17 ± 1.77 ^a^	8.00 ± 1.00 ^a^	7.50 ± 1.71 ^a^
	10	21.13 ± 2.91 ^abc^	16.63 ± 1.72 ^a^	74.27 ± 6.36 ^abc^	7.50 ± 1.12 ^bc^	7.67 ± 0.94 ^ab^	6.50 ± 1.71 ^a^	6.83 ± 2.03 ^a^	7.33 ± 1.70 ^a^	7.17 ± 1.34 ^a^
	20	23.57 ± 2.45 ^abc^	23.10 ± 2.53 ^ab^	77.87 ± 5.77 ^abcd^	7.00 ± 1.15 ^abc^	7.83 ± 0.69 ^b^	6.50 ± 1.71 ^a^	7.00 ± 1.41 ^a^	7.83 ± 1.07 ^a^	7.33 ± 1.25 ^a^
	30	14.00 ± 1.69 ^ab^	13.80 ± 1.63 ^a^	60.43 ± 4.37 ^a^	7.00 ± 0.58 ^abc^	7.83 ± 0.90 ^b^	6.67 ± 1.70 ^a^	7.00 ± 1.73 ^a^	7.33 ± 1.37 ^a^	7.33 ± 0.94 ^a^
M	0	38.37 ± 6.57 ^c^	37.50 ± 7.35 ^b^	74.57 ± 7.22 ^abc^	6.83 ± 1.34 ^abc^	7.33 ± 0.75 ^ab^	6.83 ± 0.90 ^a^	6.50 ± 1.26 ^a^	7.33 ± 0.47 ^a^	7.00 ± 0.58 ^a^
	10	31.33 ± 7.84 ^bc^	24.77 ± 4.33 ^ab^	68.00 ± 9.82 ^ab^	6.83 ± 1.30 ^abc^	6.83 ± 0.69 ^ab^	6.67 ± 0.94 ^a^	6.67 ± 1.11 ^a^	7.00 ± 0.58 ^a^	6.83 ± 1.07 ^a^
	20	30.07 ± 9.47 ^abc^	23.43 ± 9.59 ^ab^	83.03 ± 6.84 ^bcde^	6.33 ± 1.49 ^abc^	6.83 ± 1.07 ^ab^	6.33 ± 1.11 ^a^	7.00 ± 1.00 ^a^	7.50 ± 0.76 ^a^	6.83 ± 1.07 ^a^
	30	26.43 ± 6.16 ^abc^	16.83 ± 5.19 ^a^	66.20 ± 5.61 ^ab^	6.17 ± 1.46 ^abc^	6.83 ± 1.21 ^ab^	6.33 ± 1.00 ^a^	7.00 ± 1.00 ^a^	7.33 ± 1.37 ^a^	7.00 ± 1.15 ^a^
A	0	18.11 ± 4.21 ^ab^	17.20 ± 4.12 ^a^	94.93 ± 2.18 ^cde^	5.17 ± 1.21 ^abc^	6.33 ± 1.37 ^ab^	5.67 ± 0.94 ^a^	5.17 ± 2.11 ^a^	6.00 ± 2.31 ^a^	6.17 ± 0.90 ^a^
	10	14.67 ± 3.59 ^ab^	11.42 ± 2.03 ^a^	97.37 ± 6.16 ^de^	4.33 ± 1.25 ^a^	5.17 ± 1.67 ^a^	5.17 ± 1.46 ^a^	4.00 ± 1.53 ^a^	5.33 ± 2.21 ^a^	5.00 ± 1.15 ^a^
	20	17.73 ± 3.21 ^ab^	16.93 ± 3.22 ^a^	97.50 ± 7.19 ^de^	4.67 ± 1.70 ^ab^	5.50 ± 1.50 ^aba^	5.33 ± 1.60 ^a^	4.00 ± 1.15 ^a^	5.00 ± 1.63 ^a^	4.83 ± 1.77 ^a^
	30	11.73 ± 2.16 ^a^	10.95 ± 1.37 ^a^	103.20 ± 5.71 ^e^	4.50 ± 1.89 ^a^	5.33 ± 1.89 ^b^	4.67 ± 1.89 ^a^	4.33 ± 1.89 ^a^	5.00 ± 2.08 ^a^	4.83 ± 1.95 ^a^

TS—type of sample; F—crisps expanded by frying in hot oil; M—crisps expanded by microwave; A—crisps expanded by air frying; AA—additive amount (fresh carrot pulp, %); FR—fracturability; CR—crispness; H—hardness; ^a–e^—means indicated with the same letters in columns do not differ significantly at α = 0.05.

## Data Availability

The data presented in this study are available on request from the first or corresponding author.

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
