# Peer review of "Structure and Texture Characteristics of Novel Snacks Expanded by Various Methods"

_materials, 2023, doi:10.3390/ma16041541_

Round 1
Reviewer 1 Report
This work was to evaluate the structure with X-ray micro-computed tomography, texture profile, and sensory of novel potato-based snack foods supplemented with various levels of fresh carrot pulp. Three different methods of extruded snack pellets expansion were used to obtain ready-to-eat crisps: deep-fat frying, microwave and hot-air toasting. I thought this paper can be accepted after minor revision.
1. Table 1. The AA should be added the additive unit %.
2. Line 322. I did not find the 'pore size' label on the X-axis in Figure 3. But the figure 3 title noted "pore thickness". It was very confused.
3. In addition, whose wall in the Figure 3? What was the definition of wall thickness?
4. Line 325-326, again, The 'pore thickness distribution' was noted in the paper. So it was pore thickness distribution or pore size distribution?
Author Response
Reviewer 1
Comments and Suggestions for Authors
This work was to evaluate the structure with X-ray micro-computed tomography, texture profile, and sensory of novel potato-based snack foods supplemented with various levels of fresh carrot pulp. Three different methods of extruded snack pellets expansion were used to obtain ready-to-eat crisps: deep-fat frying, microwave and hot-air toasting. I thought this paper can be accepted after minor revision.
- Table 1. The AA should be added the additive unit %.
Thank you for your comment. Changes are made according to the reviewers suggestion in Table 1 and 2. Additionally, we made proper corrections in the main text and figure captions to uniform additive amount as % instead of g/100 g.
- Line 322. I did not find the 'pore size' label on the X-axis in Figure 3. But the figure 3 title noted "pore thickness". It was very confused.
- In addition, whose wall in the Figure 3? What was the definition of wall thickness?
- Line 325-326, again, The 'pore thickness distribution' was noted in the paper. So it was pore thickness distribution or pore size distribution?
Points 2-4: Thank you for this important comments. In fact, it could be misleading, these parameters should be named as “Pore size” and “Wall thickness”. Figure 3 has been replaced: the label of X-axis was supplemented, the caption of figure was corrected, and in the whole manuscript these terms were uniformed.
It was added in text in MM section: Wall thickness is the thickness of the layers of solid material located between pores and can be described as the distance between one surface of the solid and its opposite surface.

Reviewer 2 Report
The authors evaluated the microstructure of the crisp expanded by using MCT technique, instrumental texture and sensory analysis.
However, there are some issues in the text that the authors should address before publication:
1) Fig.1 it is not clear enough, all the images seams over exposed and the differences between the samples are hard to observe. I suggest the authors to change it and provide one with better contrast.
2) Fig. 3 and Fig.5 are very big and they are difficult to follow while reading the text, I suggest to put them in the supporting information
3) The small paragraph with the citation of 22, lines 285-291 ,seams about of context , it is not clear how this citation is connected to the discussion. Did the authors evaluate the Tg of the materials that they use ? In that case they should provide those data.
4) Also the citation at line 484 is out of context, since it is not connected with the discussion.
5) The authors suggest the MCT technique as a suitable method for the morphometric characterization of potatoes estrudates, but a critical discussion on the effective applicability of this technique at the industrial level is missing.
Author Response
Reviewer 2
Comments and Suggestions for Authors
The authors evaluated the microstructure of the crisp expanded by using MCT technique, instrumental texture and sensory analysis.
However, there are some issues in the text that the authors should address before publication:
1) Fig.1 it is not clear enough, all the images seams over exposed and the differences between the samples are hard to observe. I suggest the authors to change it and provide one with better contrast.
Thank you for your comment. Figure 1 was changed/improved for better contrast.
2) Fig. 3 and Fig.5 are very big and they are difficult to follow while reading the text, I suggest to put them in the supporting information
Thank you for your comment. Figure 3 is the integral part of the manuscript, as it presents the distribution of the pore size and walls thickness, which is described in the text. Thus, in our opinion it should stay in the main body of the manuscript. When opened in full screen view, the figure is of good quality and can be analyzed.
3) The small paragraph with the citation of 22, lines 285-291 ,seams about of context , it is not clear how this citation is connected to the discussion. Did the authors evaluate the Tg of the materials that they use ? In that case they should provide those data.
Thank you for your comment. In the present paper authors did not conduct such research. We understand the reviewer's comments and want to drop this paragraph. However, we have added additional citations to this section to improve discussion. We hope that in present form discussion will be more understandable.
Added in the text:
In the case of 3D rice pellets, it was noticed that the porosity ranged from 79.1% (for pellets made of a blend moistened to 29% extruded at 135ËšC) to 83.5% (for pellets made of a mixture moistened to 31% extruded at 110ËšC ) during their expansion using 800 W microwaving for 50 s. However, it was noted that in statistical terms, the pellet extrusion conditions had no statistical significance in the case of porosity [22]. In the typical potato pellets expanded by deep-frying it has been observed that the inclu-sion of fiber inhibits expansion and formation of a porous structure, which is associated with the lowering amount of starch responsible for expansion process [23].
References added:
- Zambrano, Y.; Contardo, I.; Moreno, M.C.; Bouchon, P. Effect of extrusion temperature and feed moisture content on the microstructural properties of rice-flour pellets and their impact on the expanded product. Foods, 2022, 11, 198. https://doi.org/10.3390/foods11020198
- Kita, A.; Pęksa, A.; Zieba T.; Figel, A. Influence of pellets moisture and dietary fibre addition on some potato snacks properties. Acta Agrophysica, 2002, 77, 33-43.
4) Also the citation at line 484 is out of context, since it is not connected with the discussion.
Thank you for your comment. We agree with the reviewer's comment and we rephrased mentioned sentence. Following text was added instead of previous one:
The results of statistical analysis showed that instrumental assessment of the extruded products texture has been related to sensory evaluation, as confirmed by Anton and Luciano [33], and in snack foods, texture is of major importance, with crispness being a desirable attribute.
5) The authors suggest the MCT technique as a suitable method for the morphometric characterization of potatoes estrudates, but a critical discussion on the effective applicability of this technique at the industrial level is missing.
Thank you for your comment. However, the authors in this statement referred to the validity of the morphometric characterization at the level of food science research, because as they emphasized, it is a non-invasive method of analysis (mentioned in the introduction), which is recommended in laboratory practice. On the basis of the presented results, we are unable to conduct a critical discussion on the effective applicability of this technique at an industrial level, because the application of this technique on an industrial scale requires a separate case study to prove its potential in various industries to test and compare the quality of foodstuffs. We would like to work on this topic in the future, but as a future article. Here, we only showed that the X-ray technique allows for the study of morphometric related to properties, i.e. texture evaluated instrumentally or sensorially.
